# A Novel γ′-Strengthened Nickel-Based Superalloy for Laser Powder Bed Fusion

**DOI:** 10.3390/ma13214930

**Published:** 2020-11-02

**Authors:** Jinghao Xu, Hans Gruber, Ru Lin Peng, Johan Moverare

**Affiliations:** 1Division of Engineering Materials, Department of Management and Engineering, Linköping University, SE-58183 Linköping, Sweden; ru.peng@liu.se; 2Division of Materials and Manufacture, Department of Industrial and Materials Science, Chalmers University of Technology, SE-41296 Göteborg, Sweden; hans.gruber@chalmers.se

**Keywords:** nickel-based superalloy, γ′ phase, laser powder bed fusion, alloy development

## Abstract

An experimental printable γ′-strengthened nickel-based superalloy, MAD542, is proposed. By process optimization, a crack-free component with less than 0.06% defect was achieved by laser powder bed fusion (LPBF). After post-processing by solution heat treatment, a recrystallized structure was revealed, which was also associated with the formation of annealing twins. After the aging treatment, 60–65% γ′ precipitates were obtained with a cuboidal morphology. The success of printing and post-processing the new MAD542 superalloy may give new insights into alloy design approaches for additive manufacturing.

## 1. Introduction

To improve the high-temperature mechanical and chemical properties, the nickel-based superalloy family has been developed significantly over the past several decades by chemical composition optimization. Therefore, the γ′-strengthened nickel-based superalloys are highly alloyed material systems. With the nickel acting as the austenitic matrix, it is doped with up to 10 or more alloying elements [1,2]. The addition of these alloying elements primarily serves the purpose of improving high-temperature performance. Generally, the mechanical performance is associated with the combination of the precipitation hardening effect from the ordered L1_2_ γ′ intermetallic phase and the solid solution strengthening effect from refractory elements slowly diffusing at elevated temperature. For the chemical properties, it is strongly related to the formation of protective oxide layers during high-temperature applications. Owing to the excellent high-temperature properties and wide engineering application of nickel-based superalloys, the fabrication of superalloy parts by the promising additive manufacturing (AM) techniques is of great interest. Extensive investigations have been done regarding the AM process tailoring, the post-process treatments, and characterization of high-temperature mechanical and chemical properties of a group of γ′ precipitate-strengthened nickel-based superalloys, such as IN939 [3], IN738LC [4,5,6], and CM247LC [7,8]. The compositions of these superalloys were proposed decades ago but, unfortunately, they were not developed with the intention of being adopted for AM processes.

Based on this, the need for novel chemical compositions of nickel-based superalloys ready for AM processing is urgent. However, the AM processing of precipitation-strengthened nickel-based superalloys is a great challenge, owing to its intrinsic cracking susceptibility during the AM process and/or the associated post-processing treatment. Four critical cracking mechanisms of precipitate strengthened nickel-based superalloys summarized from the welding literature below are generally accepted:
Solidification cracking (SC). During the last stage of solidification, the residual tensile stress caused by the shrinkage strain may tear apart the remaining liquid at the interdendritic region [9].Liquation cracking (LC). During the intrinsic re-heating [10] of the heat-affected zone, the (sub)grain boundaries or interdendritic regions, where the solidus temperatures are reduced by the elemental segregation during solidification, get liquified and pulled apart [11,12].Strain-age cracking (SAC). During the post-processing thermal treatment, while the material is exposed to an aging temperature, the formation of precipitates will reduce the ductility. Simultaneously, the un-released residual stress plus the precipitation stress could induce cracking when the strain exceeds the ductility limit [13].Ductility-dip cracking (DDC). During the post-processing thermal treatment of precipitate-strengthened nickel-based superalloys, a certain temperature range is critical because the ductility is highly reduced (ductility dip). Similar to SAC, a high enough strain leads to cracking in the low-ductility region [14].

In a simplistic sense, SC and LC could be attributed to the poor stress resisting capacity of the interdendritic spaces. One of the natures of the AM microstructure is the cellular-dendritic structure, whose size is around a few hundred nanometers to micrometers, depending on the specific AM process. On the other hand, these cellular structures are fine dendritic structures in terms of the elemental distribution, since, even though the cooling rate for the AM process is significantly high, the micro-segregation between the dendrite/interdendritic region cannot be fully inhibited. In the highly alloyed metallic material systems, it is widely reported that some of the elements segregate to the dendritic core region in the as-fabricated condition, such as W (in a selective electron beam melted (SEBM) CMSX-4 superalloy [15] and a laser powder bed fused (LPBF) CM247LC superalloy [16]) and Fe (in a laser beam welded (LBW) IN718 superalloy [17] and LPBF 316L steel [18]). In contrast, some other elements partitioned at the interdendritic region, such as Mo (LPBF 316L [18], LPBF IN718 [19]), Al, Ta (SEBM CMSX-4 [15], LPBF CM247LC [16]), Ti (LPBF CM247LC [16]), and Nb (LPBF IN718 [19]). Among these important major alloying elements, Al and Ti primarily act as the γ′ phase formers, while W, Ta, Mo, and Nb are usually considered the solid solution strengtheners. On this basis, we considered Mo, Nb, and Ta as the key elements that may reduce the cracking susceptibility, because (1) they are prone to segregating at the interdendritic region and could compensate for the solidification shrinkage, (2) they will enhance the strength of the interdendritic region in the solidified condition and thereby increase the cracking resistance by solid solutioning, and (3) they do not serve as the major γ′ formers.

In addition, the SAC and DDC mechanisms could simply be described as having a lack of stress releasing capacity during the post-processing thermal treatment. A sluggish recrystallization behavior is commonly found in the LPBF alloys, partially because that dislocation recovery barely happens in nickel-based superalloy systems [20,21]. These facts indicate that the stored energy is barely accommodated by the recovery-recrystallization-grain growth process. On this basis, to handle the stored energy by alternative approaches, such as tuning the composition, remains desirable in order to resist the post-processing cracking. Despite the reduction of stored energy by the conventional grain boundary (GB) migration process, the formation of annealing twins could also be an effective way to decrease the stored energy. Once the annealing twin boundary (TB) develops during the post-processing thermal treatment, these TBs provide adequate interfaces for accommodating dislocations for increasing ductility [22], thus the SAC and DDC susceptibility could be reduced. From the alloy composition modification perspective, lowering the stacking fault energy (SFE) of the alloy is usually an operative way to promote annealing twins. In the Re-free nickel-based superalloys, the Mo and Nb are the most effective interdendritic partitioning elements for reducing the SFE [23].

To this end, we developed a novel nickel-based superalloy, MAD542, for the LPBF process and its corresponding post-processing treatment. By adding appropriate levels of interdendritic segregating elements, i.e., 5 wt.% Mo and 2 wt.% Nb, crack-free parts could be fabricated in the as-built condition. Meanwhile, the addition of Mo and Nb decreased the SFE, which lowered the stored energy by TB formation. Extending this, the LPBF processing window of this novel superalloy was explored. Subsequently, the microstructures of as-built and post-processing heat-treated samples were validated by different characterization methods.

## 2. Materials and Methods

The chemical composition of the MAD542 superalloy is listed in Table 1. The thermodynamic equilibrium diagram was evaluated by a thermodynamic calculation software (version 2020b, Thermo-Calc Software, Stockholm, Sweden) with a TCNI10 database (see Figure A1 in the Appendix A). The raw powders (particle size 15–45 μm) for the LPBF process were supplied by Höganäs AB, Höganäs, Sweden. The micrographs of the powders used in this study are available in Figure A2 in the Appendix A. The LPBF process was conducted using an EOS M 100 system (EOS GmbH, Krailling, Germany) equipped with a 200 W ytterbium fiber laser source. The printing was carried out under argon atmosphere protection. To explore the LPBF processing window of MAD542, we used 9 sets of processing parameters, including 2 different laser powers, P (100 and 170 W) × 2 different scan speeds, V (1000 and 1300 mm/s) × 2 different hatching distances, and H (50 and 70 μm) + the #9 parameters with the middle values of P (135 W)/V (1150 mm/s)/H (60 μm). An identical layer thickness, L, of 20 μm was used for all the prints. The processing parameters are listed in Table 2. To compare the various processing parameters, the volume energy density, E (J/mm^3^), was calculated as E = P/(V∙H∙L) and is listed in Table 2. Nine cubes with the size of 10 mm × 10 mm × 10 mm were fabricated according to the 9 different printing parameters and with a 67° scanning vector rotation between each layer.

Metallographic sample preparation, including grinding and polishing routines, was performed. and a Leica DM6 optical microscope (OM) (Leica Microsystems GmbH, Wetzlar, Germany) was used for evaluation of AM defects. The final finish of the metallographic sample preparation was polishing with 0.04 µm colloidal silica suspension for 2 min. The defects were analyzed using the open-source image analysis software, ImageJ (1.53c, National Institutes of Health, Bethesda, MD, USA) [24]. Scanning transmission electron microscopy (STEM) was used for characterizing the as-built microstructure using a high-angle annular dark field (HAADF) detector on an FEI Tecnai G2 microscope (FEI Company, Hillsboro, OR, USA) operated at 200 kV. The thin foil was prepared by conventional twin-jet electro-polishing at −25 °C in a 10% perchloric acid and 90% ethanol electrolyte solution. The γ′ morphology was observed using a Hitachi SU70 field emission scanning electron microscope (FE-SEM) (Hitachi, Ltd., Tokyo, Japan). Electron backscatter diffraction (EBSD) measurements were conducted on the SEM, equipped with an Oxford EBSD detector (Oxford Instruments, Oxfordshire, UK).

## 3. Results and Discussion

### 3.1. LPBF Processing Window of the MAD542 Superalloy

Figure 1 shows the results from the design of experiments (DoE) to find the LPBF processing window. Owing to the various processing parameters applied for printing, an energy density study is well adapted to understand the parametric influence and reduce the complexity of the parameters [25]. By metallographic observation of the well-polished samples via OM, the level of defects could be quantitatively measured with the assistance of image analysis, e.g., threshold adjustment of binary images to identify defects. In the present study, the total level of defects was presented as the area fraction, including all cracks, porosities, and other defects. As plotted in the chart, the minimum defect level (e.g., see micrograph of #3) was achieved around 0.06% at the valley of the ‘defect area fraction vs. energy density’ curve. It suggests that the optimal energy density for printing MAD542 was between 70–80 J/mm^3^. The MAD542 superalloy printed by the optimized process parameters had good quality with no microcrack presence in the mm-scale (see Figure A3 in the Appendix A). Similar to other studies [26,27], lower energy density resulted in defects, such as porosities or ‘lack-of-fusion’ (e.g., see micrograph of #7), while higher energy density lead to cracking (e.g., see micrograph of #2). To perform the microstructural validation of the LPBF MAD542 superalloy, the #3 printing parameters, with the combination of laser power of 100 W, scan speed of 1300 mm/s, hatching distance of 50 μm, and layer thickness of 20 μm, was used for fabricating the as-built MAD542 sample.

### 3.2. As-Built Microstructure

Figure 2 shows the as-built microstructure (printed with #3 parameters in Table 2), imaged via dark-field STEM (DF-STEM) micrographs, and the elemental distribution was imaged via STEM-energy dispersive x-ray spectroscopy (EDS) composition mapping. It is well known that the cellular structure is one of the core features of the as-AM-fabricated microstructure. By measuring the center-to-center spacing (shown as the superposed networks in Figure 2a) of neighboring cells, the average cellular size was determined as 420 nm. An enlarged view of the cellular structure is provided in Figure 2b. The cellular walls were composed of dense, entangled dislocations (as indicated by the red arrow), while in the interior region of the cell, less and more sparsely distributed dislocations were observed (as indicated by the blue arrow). It should be noted that these cellular structures still represented the features of the solidification dendrites without the development of secondary dendrites. In fact, it can be supposed that the microsegregation of elements could not be fully suppressed in the LPBF process. Here, we applied STEM-EDS composition mapping of a dendrite and its surrounding region, as illustrated by the dashed box drawn in Figure 2b. Among the major alloying elements, Nb, Mo, Ta, and Ti strongly partitioned to the interdendritic region. The segregation behaviors of these elements follow the same tendencies as other LPBF nickel-based superalloys [8,28,29,30] and their conventional cast counterparts [31,32,33].

### 3.3. Heat Treatment of MAD542 Superalloy

Figure 3a–c depicts the microstructure of the LPBF MAD542 superalloy after post-processing heat treatment. For comparison, the microstructures of the heat-treated LPBF-processed γ′-strengthened superalloys IN738LC (Ni–16.2Cr–8.5Co–3.5Al–3.5Ti–2.4W–1.8Mo–1.7Ta–1Nb–0.1C–0.01B, wt.%) and CM247LC (Ni–8Cr–9.3Co–5.6Al–0.7Ti–9.5W–0.5Mo–3.2Ta–1.4Hf–0.07C–0.015B, wt.%) are illustrated in Figure 3d–i, respectively. It should be noted that IN738LC and CM247LC are two common superalloys of good interest for the LPBF process. In this study, the MAD542 was solutioning treated at 1230 °C for 2 h, the IN738LC was hot isostatic pressed at 1210 °C for 4 h (for healing the cracks), followed by solutioning heat treatment at 1120 °C for 2 h and aging at 850 °C for 24 h [34], and CM247LC was solutioning treated at 1260 °C for 2 h. It should be noted that all the three superalloys were thermal treated at the super-solvus temperature, which indicate that single γ phase region was achieved. The grains with a grain orientation spread (GOS) value less than 1° were defined as the recrystallized grains. All three samples were recrystallized (RX) according to the GOS maps shown in Figure 3a,d,g. The inverse pole figure (IPF) coloring maps, with GBs highlighted in black lines, shown in Figure 3b,e,h, revealed the grain size (GS) of the RX samples. The GBs and TBs maps are plotted in Figure 3c,f,i. It is worth noting that, compared with other LPBF precipitation-strengthened nickel-based superalloys, the MAD542 alloy shows excellent RX response during post-processing heat treatment, even though the LPBF superalloys are considered difficult to thermally treat into the recrystallized condition.

Owing to the greater amount of Mo and Nb added, annealing twins were easily promoted. We found that the TB fraction of MAD542 increased from almost null (0.3%) in the as-built condition to 40% after 10 min annealing and to 57% after 30 min annealing at 1230 °C. Meanwhile, 95% RX fraction could be achieved by 60 min annealing. See Figure A4 in the Appendix A for details regarding the RX and TB fraction evolution. As introduced above, the precipitation-strengthened nickel-based superalloys not only faced the challenging cracking issues during AM-processing, but also during the post-processing heat treatment. With the assistance of twin formation, the developed TB broke the parent GBs into finer interfacial networks. Consequently, the grain size was finer and the GS distribution was more uniform in MAD542 (Figure 3b) compared to IN738LC (Figure 3e) and CM247LC (Figure 3h). Furthermore, these interfacial boundaries effectively accommodated either the internal strains caused by residual stresses or the precipitation strains from the γ′ formation during the post-processing heat treatment. For verification, the resultant sample of MAD542 after heat treatment was still in good quality with no microcrack presence (micrograph available in Figure A5 in the Appendix A).

### 3.4. γ′ Precipitate Characterization

Figure 4 shows the micrographs of γ′ precipitates after solutioning plus aging heat treatment. In the γ′ strengthened nickel-based superalloy, the volume fraction and morphology of γ′ precipitates were well associated with the critical mechanical properties at the elevated temperature, such as creep resistance and yield strength. To reveal the grain boundary γ′ (Figure 4a,b), deep etching was conducted using electro-etching at 5 V in a 10% phosphoric acid solution. Stereo-images were generated by image pairs from ±10° angle of reproduction to illustrate the depth information, along with GB networks. In Figure 4b, high density of γ′ phases decorated the GB. The morphology and fraction of γ′ at GBs were very similar to those in the bulk grains, indicating a uniform distribution of precipitation throughout the whole sample. The γ′ precipitates in the grain interior are shown in Figure 4c,d. Cuboidal shaped γ′ were obviously identified, and the average length was measured as 372 (±76) nm. By applying image analysis, the volume fraction of γ′ in this heat treatment condition was measured to be between 60–65%. It is worth noting that after the successful precipitation of γ′, no apparent defects like cracks were introduced (see Figure A6 in the Appendix A).

## 4. Conclusions

To summarize, in this work, a novel chemical recipe of a precipitation-strengthened nickel-based superalloy, MAD542, for AM processes was proposed. By searching the processing window using a DoE matrix containing nine different printing parameters, the MAD542 superalloy was successfully fabricated in a crack-free condition with limited defects by LPBF. Beyond the excellent manufacturability, this superalloy was successfully heat-treated without any cracking occurrence. The key insight of developing the ‘non-weldable’ nickel-based superalloy in this study was to add a higher amount of Mo, Nb, and Ta. These elements segregated to the interdendritic region and helped to mitigate the interdendritic-like cracking (solidification and liquation cracking) during processing. Meanwhile, the elements Mo and Nb acted as effective SFE reducers, initiating the formation of annealing twins, which decreased the cracking susceptibility (SAC and DDC) during the post-processing treatment. The results from this study will hopefully contribute to a new paradigm for alloy design and lead to more precipitation-strengthened superalloys specifically tailored for AM processes in the future.

## Figures and Tables

**Figure 1 materials-13-04930-f001:**
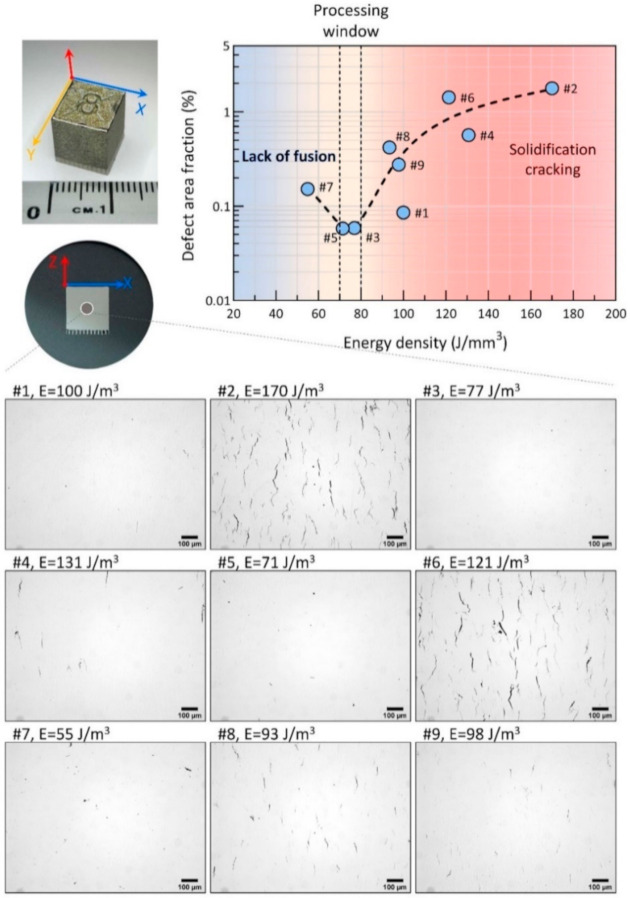
Laser powder bed fusion (LPBF) processing window investigation of MAD542 superalloy on 10 mm × 10 mm × 10 mm cubes. With lower energy density input, the major defects are dominated by lack of fusion, while higher energy density inputs result in micro-cracks. The crack-free part is provided by #3 printing parameters, which has a suitable energy density input for MAD542.

**Figure 2 materials-13-04930-f002:**
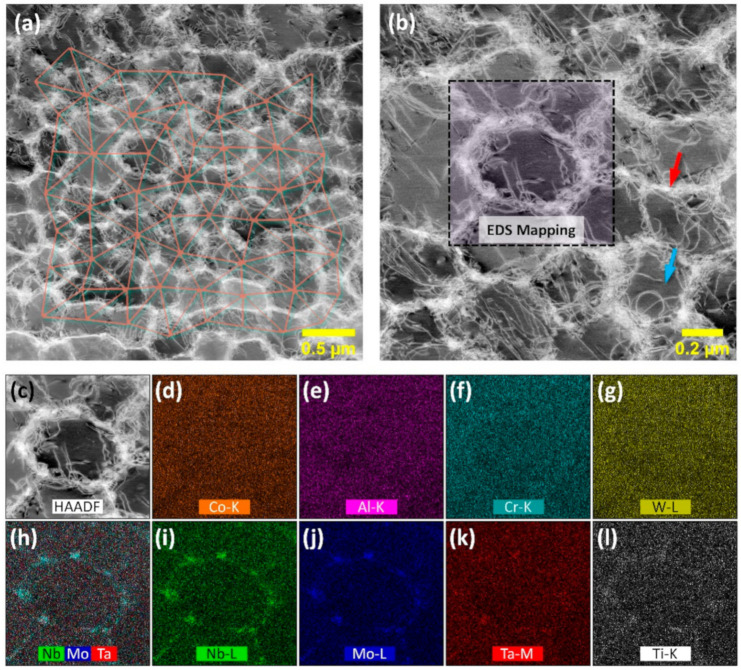
(**a**) Dark-field scanning transmission electron microscopy (DF-STEM) micrograph of the as-built MAD542 microstructure overlapped with cellular/dendritic arm spacing measurements in sub-micron size. (**b**) Enlarged view of the cellular/dendritic structure. (**c**–**l**) EDS mapping results of the scanning area as shown as the box in (**b**): (**c**) High-angle annular dark-field (HAADF) imaging, (**d**) Co-K map, (**e**) Al-K map, (**f**) Cr-K map, (**g**) W-L map, (**h**) Nb-, Mo-, Ta-overlapping map, (**i**) Nb-L map, (**j**) Mo-L map, (**k**) Ta-M map, and (**l**) Ti-K map.

**Figure 3 materials-13-04930-f003:**
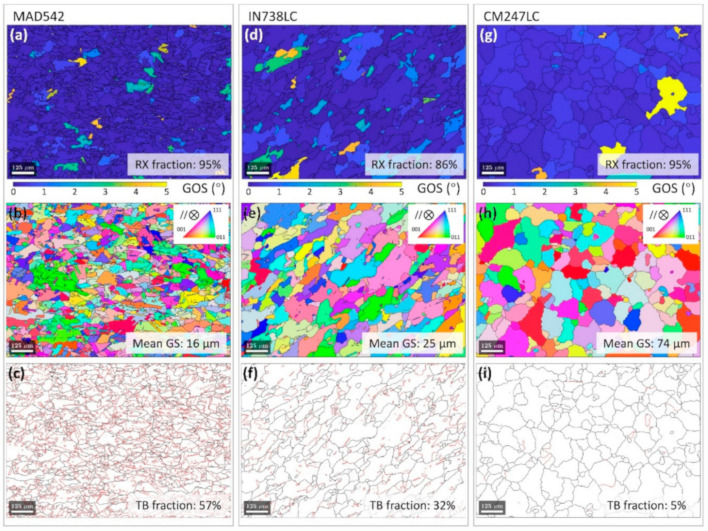
Post-processing heat treated microstructures of LPBF MAD542 (**a**) grain orientation spread (GOS) map, (**b**) inverse pole figure (IPF) coloring map, (**c**) grain- and twin-boundary map; LPBF IN738LC [34] (**d**) grain orientation spread (GOS) map, (**e**) inverse pole figure (IPF) coloring map, (**f**) grain- and twin-boundary map; LPBF CM247LC (**g**) grain orientation spread (GOS) map, (**h**) inverse pole figure (IPF) coloring map, (**i**) grain- and twin-boundary map. Electron backscatter diffraction (EBSD) step size: 2 μm.

**Figure 4 materials-13-04930-f004:**
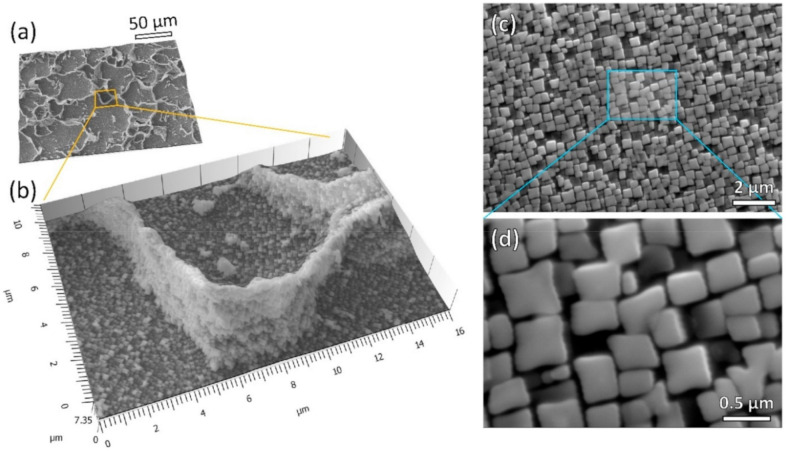
The γ′ precipitate morphology of MAD542 superalloy after post-processing heat treatment. (**a**) Stereo-image from SEM-secondary electron (SE) micrographs on a deep etched sample, showing a high amount of γ′ precipitate decorating the grain boundary region with an enlarged view in (**b**); (**c**) SEM-SE imaging of high density of γ′ precipitates observed in the grain interior region; (**d**) the enlarged view of cuboidal γ′ morphology.

**Table 1 materials-13-04930-t001:** The chemical composition of the MAD542 nickel-based superalloy investigated in this study.

Element	Cr	Co	Mo	W	Al	Ti	Ta	Nb
wt.%	8	8	5	4	5	1	3	2
**Element**	**C**	**B**	**Si**	**P**	**S**	**Zr**	**O**	**Ni**
wt.%	0.1	<0.001	<0.005	<0.001	<0.001	<0.002	0.015	Bal.

**Table 2 materials-13-04930-t002:** Laser powder bed fusion processing parameters used in this study (layer thickness: 20 μm).

Exp	Laser Power, P (W)	Scan Speed, V (mm/s)	Hatching Distance, H (μm)	Energy Density, E (J/mm^3^)
#1	100	1000	50	100
#2	170	1000	50	170
#3	100	1300	50	77
#4	170	1300	50	131
#5	100	1000	70	71
#6	170	1000	70	121
#7	100	1300	70	55
#8	170	1300	70	93
#9	135	1150	60	98

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
