# Peer review of "A Novel γ′-Strengthened Nickel-Based Superalloy for Laser Powder Bed Fusion"

_materials, 2020, doi:10.3390/ma13214930_

Round 1
Reviewer 1 Report
The manuscript is well written and the results will be a valuable contribution to the literature on the additive manufacturing of superalloys.
Author Response
Please see the attachment.
Regards,
Jinghao Xu, Hans Gruber, Ru Lin Peng, Johan Moverare

Reviewer 2 Report
The paper entitled "A novel γ′-strengthened nickel-based superalloy for laser powder bed fusion" presents high quality research about additive technology applied to design new alloys for 3D printing. The paper is important because the pallet of materials for 3D technology is very limited. The experiment is well prepared and described. The results are promising and it gives another point for knowledge to materials and technology design.
The paper may be published after minor correction. I found some shortcomings:
1. Line 27: L_12 -> L1_2
2. Line 115: The authors mentioned imageJ software used in the paper. Please add the citation according to https://imagej.nih.gov/ij/docs/faqs.html
3. Caption of Fig. 3: The authors used shortcuts like GOS, IPF, but the meaning is in the text. It is better to include the meaning in the caption, as is in Fig. A4.
Author Response

(The authors gave the same response as above.)

Reviewer 3 Report
The paper presents the results of a valuable characterization of an experimental printable Nickel-based superalloy.
However, it should be improved for the following points:
- The chemical composition reported in Table 1 is merely a nominal one. Table 1 should include the specified composition limits defined by the authors. Moreover, the powder analysis of the product supplied by Höganas should be reported, as well as the product analysis of the printed product, to be compared with the specified ranges.
- Section 3 and Figure 1 discuss parameters allowing to produce crack-free parts, with reference to Table 2. However, it is not explicitly said in the text that n° 3 parameters are retained. It is only mentioned a range of energy density between 70 J/mm3 and 80 J/mm3 that could correspond as well to parameter n° 5. In particular, it is not clear to which parameters corresponds the sample studied in Fig. 2. The reader has to refer to the legend of Fig. A3 to guess that it is n° 3 parameters that are retained (and applicable to the results further reported in the paper?).
- Section 3 includes a comparison between LPBF MAD542 and two other alloys (IN738LC and CM247LC). The chemical composition of these two alloys should be provided as well. Moreover, the comparison is not really consistent since it includes an alloy IN738LC which is the only one having undergone a HIPing cycle. It should be critically discussed why a HIPing cycle was necessary and applied to this alloy, but not to the one studied by the authors and the second alloy used for comparison (CM247LC)
- It is said with reference to Fig. A5 that MAD542 after heat treatment is still in "good quality", however it should be clarified the meaning of "good" (absence of cracking?). Moreover, it should be clarified how the specimens were prepared for the observations reported in Fig. A5 and A6 (simple polishing?)
- It is also suggested that the authors apply non-destructive testing (namely Penetrant Testing) in order to further check for the absence of porosity and/or linear imperfections in the samples obtained with the optimized parameters
- Finally, spelling should be checked (disusing for diffusing, w for W) etc.
Author Response

(The authors gave the same response as above.)

Reviewer 4 Report
This interesting paper deals with the gamma prime strengthened of nickel based super alloys. The objectives is to obtain crack free material by optimize the fabrication process with respect to the heat treatment. Nevertheless, some points need to be highlighted. Some comments were done as followed in order to improve the quality of the paper:
1) The abstract is good, but Authors need to give results found very directly and clearly at the very end.
2) Authors need to improve the very end of introduction to differentiate the work done from the literature study. It could be helpful to describe the plan of the paper to guide the reader.
3) In “Materials and Methods”, various thinks are missing
- a) The method need to be more described chronologically, in order to highlight the different steps, such as: the fabrication, the heat treatment, the polishing methods, the measurement and the post-treatment.
- b) The heat treatment for the three different materials need to be explained in this section and why such heat treatment.
- c) Number of samples are missing.
- d) Fig A1 and A2 mentioned in this section need to be presented and discussed, if not why showing them?
- e) Selection of DOE design and range of parameters should be discussed
4) “Results and Discussion” section needs to be more structured, and some subsections need to be created.
In this section the “sparsely distributed dislocations” have to be pointed in the figure 2b)
5) It is hard to evaluate the quality of the Figures A3, A5 and A6. Are they blurred or presented with free-default. So why showing them?
Author Response

(The authors gave the same response as above.)

Reviewer 5 Report
The manuscript presents an investigation in additive manufacturing, using PBF of a new nickel alloy.
The study is contemporary and of great interest to the academic community.
The manuscript is well structured and well written.
It would be beneficial to outline in the manuscript the expected properties of the printed material as well as potential uses of the newly developed alloy.
Author Response

(The authors gave the same response as above.)

Reviewer 6 Report
The present work involves the development of a new Ni-based superalloy designed for additive manufacturing, with a particular focus on the reduction in the susceptibility of the material to solidifaction cracking. It is argued that addition of certain refractory alloying elements, typically understood to be solid-solution strengtheners, will serve this purpose because of their tendency to segregate to the interdendritic region and reduce the stacking-fault energy (increasing annealing twin boundary formation).
Powders containing addition of some refractory elements are procured and an empirical processing study is performed with accompanying characterization. This reviewer has only minor comments.
1. There was (presumably) some design work necessary to determine the alloy composition, but the rationale for the specific alloying additions made is not discussed.
2. While the introduction contains a lot of great fundamental metallurgy, this section is meandering and confusing. Much of the review content could be removed to improve readability and the section refocused on the benefits of refractory alloying additions and closely-related work.
3. The rest of the paper is pretty clear. The experimental results are consistent with the authors' hypothesis regarding refractory additions and twin boundary formation.
Author Response

(The authors gave the same response as above.)

Round 2
Reviewer 4 Report
Corrections from reviewer have been seriously taken into consideration.